# Emerging Roles and Potential Applications of Non-Coding RNAs in Glioblastoma

**DOI:** 10.3390/ijms21072611

**Published:** 2020-04-09

**Authors:** Carlos DeOcesano-Pereira, Raquel A. C. Machado, Ana Marisa Chudzinski-Tavassi, Mari Cleide Sogayar

**Affiliations:** 1Center of Excellence in New Target Discovery (CENTD), Butantan Institute, 1500 Vital Brazil Avenue, São Paulo 05503-900 SP, Brazil; carlos.ocesano@butantan.gov.br (C.D.-P.); ana.chudzinski@butantan.gov.br (A.M.C.-T.); 2Department of Life Science and Medicine, University of Luxembourg, Campus Belval, Avenue des Hauts-Fourneaux, L-4362 Esch-sur-Alzette, Luxembourg; rmachado@iq.usp.br; 3Biochemistry Department, Chemistry Institute, University of São Paulo, São Paulo 05508-000, Brazil; 4Cell and Molecular Therapy Center (NUCEL), School of Medicine, University of São Paulo, São Paulo 05360-130 SP, Brazil

**Keywords:** glioblastoma, non-coding RNAs, miRNAs, lncRNAs, regulation of gene expression

## Abstract

Non-coding RNAs (ncRNAs) comprise a diversity of RNA species, which do not have the potential to encode proteins. Non-coding RNAs include two classes of RNAs, namely: short regulatory ncRNAs and long non-coding RNAs (lncRNAs). The short regulatory RNAs, containing up to 200 nucleotides, include small RNAs, such as microRNAs (miRNA), short interfering RNAs (siRNAs), piwi-interacting RNAs (piRNAs), and small nucleolar RNAs (snoRNAs). The lncRNAs include long antisense RNAs and long intergenic RNAs (lincRNAs). Non-coding RNAs have been implicated as master regulators of several biological processes, their expression being strictly regulated under physiological conditions. In recent years, particularly in the last decade, substantial effort has been made to investigate the function of ncRNAs in several human diseases, including cancer. Glioblastoma is the most common and aggressive type of brain cancer in adults, with deregulated expression of small and long ncRNAs having been implicated in onset, progression, invasiveness, and recurrence of this tumor. The aim of this review is to guide the reader through important aspects of miRNA and lncRNA biology, focusing on the molecular mechanism associated with the progression of this highly malignant cancer type.

## 1. Introduction

For decades, it has been believed that the central dogma of Molecular Biology (DNA → mRNA → Protein) was unidirectional, with all the complex cellular processes of an organism being solely due to the structural and catalytic functions of proteins [1]. Currently, it is widely accepted that this biological complexity is derived from the non-coding DNA portion of the genome, which was once thought of as ‘junk DNA’ (non-codifying DNA). Non-coding RNA (ncRNA) is a class of RNAs which has no potential for translation into proteins. The DNA sequence from which a functional ncRNA is transcribed is often called an RNA gene. The number of ncRNAs within the human genome is unknown; however, massive expansion of global transcriptome datasets from genomics consortia have been demonstrating that most of the human genome is transcribed into non-coding RNAs [2,3,4]. According to the Encyclopedia of DNA elements (ENCODE) project, approximately 75% of the human genome is actively transcribed into ncRNAs [2,5], only 2% of which code for known protein-coding genes [5]. Importantly, ncRNAs have been revealed to be functional and to form complex regulatory networks associated with several biological processes. Disruption of key components of these networks leads to deregulated cell function and contributes to human disease states, including cancer [6,7,8]. 

Glioblastoma is the most aggressive type of brain cancer in adults, accounting for about half of all primary brain tumors [9]. Despite the multimodal treatment procedure, which consists of maximal resection followed by radiotherapy and chemotherapy, the overall survival rate remains only 12–15 months, highlighting the urgent need for more effective targeted therapy [10]. 

New insights into the molecular subtypes of diffuse gliomas have led to unprecedented discoveries of potential prognostic and predictive markers [11]. The latest classification system announced by the World Health Organization (WHO) [12] combines the classical histomorphological analysis with molecular genetic tests, allowing more precise diagnosis and guidance for therapeutic interventions [13]. Further studies in this direction should provide the basis for the development of novel therapeutic strategies targeting unique molecular signatures for patient-tailored treatment.

Non-coding RNAs have increasingly been described as biomarkers of various human diseases [14,15,16,17,18] and/or suggested as therapeutic targets [19,20,21,22,23]. In this context, the aim of this article is to review the exciting progress towards elucidating the multifunctional facet of ncRNAs, with special focus on glioblastoma-associated miRNAs and lncRNAs. Finally, we also discuss the limitations and obstacles to translate these findings into the clinical practice.

## 2. Classification and Biogenesis of ncRNAs

### 2.1. Classification

The classification of ncRNAs is based on their structure, function, biogenesis, localization, and interaction with DNA or protein-coding mRNAs [24,25]. The short regulatory RNAs, including small RNAs, have up to 200 nucleotides, whereas RNAs with more than 200 nucleotides are called long non-coding RNAs (lncRNAs). Abundant and functionally important types of small non-coding RNAs include transfer RNAs (tRNAs) and ribosomal RNAs (rRNAs), as well as small RNAs, such as microRNAs (miRNAs), small interfering RNAs (siRNAs), piwi-interacting RNAs (piRNAs), small nucleolar RNAs (snoRNAs), small nuclear RNAs (snRNAs), extracellular RNAs (exRNAs), and small Cajal body-specific RNAs (scaRNAs) [24,25,26,27]. The proportion of different ncRNAs to the total amount of RNA in eukaryotic cells ranges between 0.002–0.2%, while rRNA and tRNA represent 80–90% and 10–15%, respectively [28]. The lncRNAs are classified into antisense lncRNAs [29] and long intergenic ncRNAs (lincRNAs) [30], depending on the region of transcription of these RNAs, with respect to the protein-coding genes located on the same genomic locus. Natural antisense transcripts (NATs) constitute a category of lncRNAs which exhibit sequence-complementarity to other RNA molecules [31]. NAT is a non-protein-coding antisense RNA partner to a protein-coding mRNA, indicating that it might also regulate non-protein-coding sense RNAs [31]. 

### 2.2. Biogenesis

The biogenesis of ncRNAs is based on their properties, which are similar to those of mRNAs. Many ncRNAs share structural properties with protein-coding mRNAs, such as alternative splicing, 5′-cap modification and 3′-polyadenylation [32,33], sequence conservation [32] and transport of RNA molecules from the nucleus to the cytoplasm [34,35]. On the other hand, some ncRNAs do not share these similarities, highlighting the existence of differences between coding versus non-coding RNAs [36,37]. Therefore, each type of RNA has specific checkpoints in expression and different biological roles. Many lncRNAs are located exclusively in the nucleus, and others are cytoplasmic or are located in both nucleus and cytoplasm [33].

MiRNAs are transcribed from the DNA genome into primary miRNAs (pri-miRNAs) and post-transcriptionally processed into ~70 nucleotides hairpin precursor miRNAs (pre-miRNAs) and, finally, to ~22 nucleotides mature miRNAs [38]. In general, miRNA biogenesis starts with the processing post- or co-transcriptionally of transcripts by RNA Polymerase II (RNAP II) [39]. MiRNAs processing begins in the nucleus under the action of Drosha and ends in the cytoplasm upon the activity of the Dicer enzyme, where they are accumulated to interact with protein-coding mRNAs [40]. Depending on the physiological cellular condition, some miRNAs may be secreted into extracellular vesicles and transported as exosomes to other cells [38].

LncRNAs are transcribed under the action of RNAP II [33,41], similarly to protein coding-mRNA, but are usually devoid of open reading frames (ORFs) [42]. Most long transcripts with known non-coding functions contain multiple ORFs in their sequences. These ORFs may not be translated, or be inefficiently translated or translated to produce an aberrant (truncated) protein, which has no functional consequences, since it is rapidly degraded intracellularly. LncRNAs may or may not contain the 3′ poly-A tail [28,42,43]. Polyadenylated and capped RNAP II-transcribed mRNAs are similar to lncRNA, however, a fraction of the lncRNAs transcribed by RNAP II may lack structural regions (lack of a 5′-cap structure and/or of a 3′poly-A tail) and may undergo a different 3′-end processing [44,45]. Interestingly, expression of lncRNA is more tissue-specific than that of mRNA, suggesting that their transcription is highly regulated [46,47].

Since lncRNAs are master regulators among a plethora of different cellular processes, control of their metabolism and biogenesis is crucial for modulation of their biological function. This control of biological pathways occurs at both transcriptional and post-transcriptional levels, including: fine-tuning alternative splicing, mRNA stability, scaffolding, gene expression regulation, chromatin remodeling (histone modification and DNA methylation), nuclear trafficking, and mRNA translation [48,49]. The rate of synthesis and degradation of these molecules is critical for several important biological conditions and deregulation of these processes may contribute to the onset of cancer [50].

## 3. Functional Roles and Mechanisms of Action of ncRNAs

### 3.1. Biological Function

Functional ncRNAs have been identified as important players in the development of diseases and several studies have highlighted how regulatory RNAs play crucial roles in cancer onset and progression [26,51]. Several families of ncRNAs perform a wide range of functions, which lead to modulation of transcription and to control of synthesis and/or activity of specific proteins; as well as binding to specific regions of DNA, culminating in activation or repression of transcription [48]. Although a large number of ncRNAs have already been documented in the literature, only a few of them have had their biological functions experimentally validated. These ncRNAs are involved in basic processes of gene regulation [51], including:
(i)Direct transcriptional regulation;(ii)Regulation of RNA processing events such as splicing, editing, subcellular localization, translation, and turnover/degradation;(iii)Chromatin modification;(iv)Regulation of genomic imprinting;(v)Post-translational regulation of protein activity;(vi)Facilitation of ribonucleoprotein (RNP) complex formation;(vii)Modulation of miRNA expression;(viii)Gene silencing through production of endogenous siRNA (endo-siRNA).


### 3.2. Mechanisms of Action

MiRNAs and lncRNAs are structurally similar and both play crucial roles in modulation of gene expression (Figure 1). Mechanistically, most miRNAs interact with the 3′-UTR or the 5′-UTR of target mRNAs to suppress gene expression or to block protein translation, respectively. In some cases, miRNAs also interact with gene promoters through specific proteins, leading to transcriptional control of the activation or repression of gene expression [38]. 

LncRNAs display various types of general mechanisms of action, which can be categorized according to their functions, and may be defined as one or more of the following archetypes [52]:(i)Signal: functions as a molecular signal inducing transcriptional activity. LncRNAs display tissue-specific expression and respond to different cellular stimuli, highlighting that their expression is highly controlled at the transcriptional level. They may act as molecular signals, since their transcription occurs in an orchestrated manner and depends on their subcellular location, allowing integration of responses to the different stimuli received;(ii)Decoy: binds to and titrates away other regulatory RNAs (e.g., miRNAs) or proteins (e.g., transcription factors). LncRNAs can act as a molecular sponge for RNA-binding proteins, such as chromatin remodelers, transcription factors or other regulatory factors. This mechanism plays a central role in both positive and negative transcription regulation by lncRNAs;(iii)Guide: directs the localization of ribonucleoprotein (RNP) complexes to specific targets (e.g., chromatin modification enzymes are recruited to promoter regions of the DNA). LncRNAs act as guides, directly binding to proteins and thus altering the location of RNPs to target regions, leading to changes in the pattern of gene expression. The regulatory components triggered by the lncRNAs include both repressive and activating complexes, as well as transcription factors;(iv)Scaffold: has a structural role as a platform upon which relevant molecular components (proteins and/or RNA) can be assembled into a complex. Molecular components can be assembled on lncRNAs, which can thus serve as a central platform, which will lead to transcriptional activation or repression. LncRNAs may bind to multiple effector partners forming complex scaffolding. In this network of interactions, these lncRNAs are responsible for addressing proteins to specific sites in the genome, thereby activating or repressing gene expression;(v)Enhancer: controls higher-order chromosomal looping in an enhancer-like model. In this functional archetype context, the levels of enhancer ncRNA (eRNA) positively correlate with the levels of messenger RNA synthesis, which are regulated by these lncRNAs, suggesting a ‘promoter-like’ role in gene expression control. The eRNAs act by recruiting the RNAP II to the promoter region.


## 4. Expression and Function of ncRNAs in Glioblastoma

Due to their biological relevance, a growing number of ncRNAs has been associated with the pathophysiology of different human diseases. In every major cancer type, disruption of ncRNA regulatory networks has already been described to have oncogenic or tumor-suppressive effects [53]. Likewise, in glioblastoma, miRNAs and lncRNAs have been implicated in all aspects of tumor malignancy, including: cell proliferation, resistance to apoptosis, metastasis, angiogenesis, and drug resistance. In the following sections, we summarize selected miRNAs and lncRNAs, which are of particular interest, focusing on the specific mechanisms of action by which these molecules might be linked to clinical manifestations of this highly malignant cancer type.

### 4.1. Deregulated miRNAs in Glioblastoma Onset/Progression

Expression profile studies have revealed that 256 miRNAs are found to be upregulated in glioblastoma and 95 to be significantly downregulated, when compared to the normal brain tissue [54]. Several studies have also identified miRNA signatures, which may be used as a function of patient prognosis and survival [55]. MiRNA signatures have also been recently associated with glioblastoma-stem cells and radiotherapy resistance [56,57]. Yet, only a small proportion of these miRNAs have been functionally described to date. A single miRNA may target several mRNAs, thus interfering in different pathways, which often modulate cell proliferation, invasiveness, programmed cell death, and angiogenesis. Here, we discuss, in detail, the most well-characterized examples linked to glioblastoma progression and/or patient survival. Finally, we summarize in Table 1 (oncogenic) and Table 2 (tumor suppressor) the main oncogenic and tumor suppressor miRNAs, with their respective validated targets and functional characteristics which are relevant for glioblastoma pathogenesis. 

#### 4.1.1. Oncogenic miRNAs

##### *hsa-miR-21* 

As suggested by Moller et al., over 70% of the miRNAs found to be deregulated in glioblastoma, are upregulated in the tumor tissue, when compared to the normal brain tissue [54]. Mir-21 is the most extensively investigated miRNA, being consistently reported to be overexpressed in different cancer types [58]. It was also the first oncogenic miRNA associated with glioblastoma, being initially described as an anti-apoptotic and pro-survival factor [59]. In fact, *miR-21* is expressed in a grade-specific manner in gliomas [59] and is inversely associated with patient survival in glioblastoma patients [60]. 

Numerous miR-21 targets have been validated in glioblastoma, with a direct impact on different aspects of tumor pathogenesis. For instance, mir-21 was shown to affect apoptosis and cell cycle by directly targeting heterogeneous nuclear ribonucleoprotein K (HNPRK), the tumor suppressor homolog of p53 (TAP63), programmed cell death 4 (PDCD4), and other targets of p53 or TGF-β signaling or the mitochondrial apoptotic pathway [61,62]. Therefore, by targeting multiple molecules, miR-21 strongly reduces the effects of chemo- and radiotherapy [63,64] and miR-21 inhibition greatly enhances the chemo-sensitivity of glioblastoma cells to different drugs, including temozolomide (TMZ), carmustine (BCNU), and sunitinib [65,66,67]. Moreover, miR-21 also promotes glioblastoma invasiveness through suppression of the expression of matrix metalloprotease (MMP) inhibitors, such as *TIMP3* and *RECK* [68]. Together, these data show that deregulated expression of *miR-21* affects various molecular targets, which may contribute to glioblastoma initiation and progression.

##### *hsa-miR-10b* 

MiR-10b was identified as an unique miRNA, which is specifically expressed in glioma tumors, but not in normal brain tissues, representing a rare unifying element, since it occurs in at least 90% of all glioblastoma subtypes [69]. Moreover, *miR-10b* expression was shown to be directly associated with gliomas pathological and malignancy grade, significantly correlating with poor prognosis in glioblastoma patients [69,70,71].

Mir-10b was shown to regulate glioblastoma invasive growth by repressing *HOXD10* expression and indirectly modulating MMP14 and uPAR [71]. Silencing of miR-10b reduced cell proliferation, invasion and angiogenesis, while cell death was increased [70,71,72]. In an orthotopic human glioma mouse model, inhibition of miR-10b significantly prolonged survival of tumor-bearing mice as a result of increased expression of multiple tumor suppressors [72]. MiR-10b also regulates cell cycle and alternative splicing through non-canonical targeting via 5’-UTRs of its target genes, including: MBNL1-3, SART3, and RSRC1 [73]. In this study, Teplyuk et al. also assessed the efficiency of different delivery protocols of mir-10b antisense oligonucleotide inhibitors in vivo [73]. The various models tested were highly responsive to anti-miR-10b therapy, including systemic injections of miR-10b inhibitor, which was sufficient to delay glioblastoma progression, did not cause toxicity and was well tolerated. Another study showed that virus-mediated miR-10b gene ablation strikingly impairs tumor growth in intracranial glioblastoma models, representing a promising therapeutic approach to be explored in the clinics [74]. These observations indicate that miR-10b functions as a pleiotropic regulator, which is essential for glioblastoma growth and survival.

##### *hsa-miR-221/222* 

*MiR-221/222* is found among the most common and significantly overexpressed miRNAs in glioblastoma. An increased level of miR-221/222 positively correlated with the degree of tumor infiltration, serving as a prognosis factor for patients [75]. Plasma miR-221/222 levels were also shown to be significantly upregulated in glioma patients and to correlate with poor survival rate, therefore, this miRNA may also be considered as prognostic biomarkers for gliomas [76].

Upregulation of *miR-221/222* promotes cell proliferation, cell cycle progression, migration, and invasion through direct inhibition of p27, p57, Cx43, tyrosine phosphatase μ (PTPμ), and TIMP3 [75,77,78,79,80]. High levels of *miR-221/222* also inhibit cell apoptosis by targeting p53-upregulated modulator of apoptosis (PUMA), which, in turn, promotes rapid cell death by binding to Bcl-2 and Bcl-xL [81]. Knockdown of miR-221/222 enhances the cytotoxic effects of TMZ treatment by regulating apoptosis and through suppression of MGMT levels in glioblastoma cells [82,83]. Overall, given its critical role in DNA damage repair, inhibition of miR-221/222 may serve as a potential strategy to treat chemo- and radioresistant glioblastomas, since it render cells unable to repair genetic damage [83]. Importantly, combined anti-miR-221/222 therapy and radiotherapy was shown to suppress tumor growth more efficiently than anti-miR-221/222 or radiotherapy alone in a murine glioblastoma model [84]. Therefore, these studies demonstrate the functional roles of the miR-221 and miR-222 paralogues in glioblastoma, suggesting potential clinical applications as a biomarker and target therapy. 

#### 4.1.2. Tumor Suppressor miRNAs

##### *hsa-miR-128* 

MiR-128 is a brain-enriched miRNA, which is found to be significantly downregulated in glioma tissues and cell lines [105,106,107]. Importantly, analysis of the TCGA database demonstrated a significant association of low expression levels of *miR-128* with high risk and poor survival rate of patients with glioblastoma [108]. Additionally, higher levels of *miR-128* expression were identified as a protective factor for *IDH* wild type glioblastomas, as part of a 23-miRNA signature [109].

Several studies have demonstrated that miR-128 exerts its function as a tumor suppressor by targeting a wide range of oncogenic protein-coding mRNAs. For instance, expression of *miR-128* was shown to decrease tumorigenesis, inhibit cancer stem cell self-renewal, and decrease radiation resistance through targeting the Bmi-1 and SUZ12 oncogenes, known to be members of the Polycomb Repressive Complex (PRC) [105,110,111]. Overexpression of *miR-128* alone also enhanced TMZ-induced apoptotic death of glioma cells through direct modulation of key members of the mammalian target of rapamycin (mTOR) signaling, including mTOR, rapamycin-insensitive companion of mTOR, insulin-like growth factor 1 (IGF-1), and phosphoinositide-3-kinase regulatory subunit 1 (PIK3R1) [108]. Moreover, overexpression of *miR-128* significantly reduced cell proliferation and also suppressed tumor growth and angiogenesis in vivo by targeting p70S6K1 and affecting its downstream signaling molecules, such as HIF-1 and VEGF [112]. Overall, these studies demonstrate the crucial role of miR-128 as a potent inhibitor of gliomagenesis.

##### *hsa-miR-181b* 

*MiR-181b* was found to be downregulated in glioma samples, when compared with normal brain tissue, and low expression of *miR-181b* was associated with poor survival of glioblastoma patients [113,114,115]. Moreover, the levels of *miR-181b* expression in high-grade gliomas (WHO clinical stage III–IV) are much lower than in low-grade gliomas (WHO clinical stage I–II), strongly suggesting that *miR-181b* expression contributes to the malignant progression of the disease [114,115]. 

MiR-181b interferes with multiple oncogenic pathways via targets involved in glioblastoma pathogenesis. Consequently, overexpression of *miR-181b* significantly decreases cell proliferation, migration, and invasion of glioblastoma cells [114,116,117] and suppresses tumor growth and angiogenesis in vivo [114]. In response to TMZ treatment, *miR-181b* expression reduced chemoresistance of glioblastoma stem cells (GSCs) promoting inhibition of secondary neurosphere and soft agar colony formation by suppressing the levels of cyclin D1, c-Myc, and Ki-67 proteins, which are essential for cell proliferation [118]. *MiR-181b* expression also conferred glioblastoma cell sensitivity to teniposide by binding to the 3′-UTR region of MDM2, leading to its reduced protein levels [119]. Therefore, miR-181b may be considered as an onco-suppressive miRNA. 

##### *hsa-miR-137* 

MiR-137 is another well-characterized miRNA with tumor-suppressive functions. Its expression is epigenetically inhibited in different types of cancer, including glioblastoma [120,121]. Li et al. also detected a significant reduction of circulating miR-137 in serum samples collected from glioblastoma patients, when compared to healthy controls [122]. Additionally, low levels of miR-137 in the tumor tissue and in serum samples were shown to be associated with a poor prognostic phenotype of glioblastoma [122].

MiR-137, as well as miR-124 have been implicated in differentiation of neural cells and GSCs [121]. Re-expression of *miR-137* in vitro can induce phenotypic changes, growth arrest, and differentiation in GSCs [120,121]. MiR-137 also plays an important role in cell cycle progression, cell motility, and apoptosis by regulating CyclinD1, MMP-2, Bcl-2, and Bax, mediated via Rac1 [123]. Furthermore, miR-137 promotes suppression of EGFR signaling in glioblastoma cells through direct targeting of EGFR [124]. In a mouse xenograft model, ectopic expression of *miR-137* inhibited tumor growth and angiogenesis by directly regulating the level of the Polycomb group proteins [122]. These results indicate that miR-137 may serve as a biomarker in glioblastoma and modulation of its activity may be a potential therapeutic strategy for treatment of this disease.

### 4.2. Deregulated LncRNAs in Glioblastoma Onset/Progression

Genome-wide studies have demonstrated that most diseases have genetic variants located outside the protein-coding genes [137]. These findings have also highlighted the potential involvement of lncRNAs in glioblastoma [138,139,140]. LncRNAs are found to be aberrantly expressed in gliomas and a lncRNAs signature has been associated with overall survival in glioblastoma patients [141]. The following section provides an accurate description of the potential functions and mechanism of action of lncRNAs in the pathogenesis and development of glioblastoma. The main lncRNAs are discussed in detail in Table 3 and Table 4, for, respectively, oncogenic and tumor suppressors, summarizing the molecular mechanisms by which these transcripts exert their functions in cells and their effects in glioblastoma pathogenesis.

#### 4.2.1. Oncogenic LncRNAs

##### *MALAT1* 

Metastasis-associated lung adenocarcinoma transcript 1 (*MALAT1*), also known as *NEAT2* (non-coding nuclear-enriched abundant transcript 2), is a long non-coding RNA subject to further processing and to post-transcriptional modifications. It is highly conserved among mammals and nuclear-retained regulatory RNA [142]. *MALAT1* was originally identified to be a highly expressed RNA in lung cancer and to be a prognostic biomarker for poor clinical outcome in patients with early-stage non-small cell lung cancer (NSCLC) [143]. Moreover, *MALAT1* has been described as highly expressed in glioblastoma, with its expression pattern corresponding to poor prognosis for glioblastoma patients [144]. Mechanistically, MALAT1 is recruited to nuclear paraspeckles and regulates pre-mRNA processing by association with SRSF2 splicing factors [145,146]. Recent studies have shown that MALAT1 acts as a competitor of cellular endogenous RNAs (ceRNAs), a class of short ncRNA, which serves as a ‘molecular sponge’ for miRNAs, leading to modulation of their downstream roles in glioblastoma [147]. Through this mechanism of action, MALAT1 blocks the activity of several miRNAs, such as miR-1066-5p, miR-144-3p, miR-211, miR-203, and miR-155 [147,148,149,150,151]. Keman Liao et al [144] showed that MALAT1 positively modulates the expression of ZHX1, acting as ceRNA by sponging miR-199a. The complex formed by MALAT1/miR-199a/ZHX1 promotes glioblastoma cell proliferation in vitro and in vivo, and correlates with glioblastoma progression in patients [144]. Consequently, MALAT1 depletion leads to inhibition of cell proliferation, promotes cell death by apoptosis and decreases invasion in vitro, in addition to reducing the tumor volume in an orthotopic murine model [144]. In glioblastoma patients under treatment with TMZ, *MALAT1* expression was upregulated, showing resistance to drug treatment, through suppression of miR-203 and activation of thymidylate synthase (TS) mRNA function [152]. Therefore, molecular therapies based on *MALAT1* knockdown could be a promising treatment for glioblastoma.

##### *MEG3* 

Maternally expressed gene 3 (*MEG3*), is a long non-coding RNA located on chromosome region 14q32.3 in humans, which belongs to the DLK1–DIO3 domain genomic locus [153]. This region contains multiple maternally and paternally imprinted genes, including three paternally expressed protein-coding mRNA and multiple maternally expressed ncRNAs [154]. *MEG3* is an imprinted gene, which is highly expressed in the human pituitary and was first identified as the ortholog for gene trap locus 2 (*Gtl2*) in mice [155]. This gene encodes a non-coding RNA of approximately 1700 nucleotides and numerous alternatively spliced transcript variants have been transcribed from this gene, all of which are lncRNAs. Experimental evidence demonstrates that this gene acts as a lncRNA tumor suppressor gene [153], however, other studies have shown that *MEG3* can also act as an oncogene [156]. *MEG3* interacts with the p53 tumor suppressor, leading to control of expression profiling of p53-target genes [153,157]. In glioblastoma, several studies indicate participation of *MEG3* in the pathogenesis and development of this disease [158,159,160]. Compared with normal brain tissue, *MEG3* was found to be downregulated in glioma and glioblastoma cell lines and overexpression of *MEG3* leads to increased cell death and inhibition of cell proliferation [159]. Other studies have shown that the *MEG3* promoter is hyper-methylated by DNMT1 in glioblastoma tissue, and, therefore, this gene is completely silenced [161]. Additionally, MEG3 acts as a ceRNA to miR-19a, which represses *PTEN* expression, leading to glioblastoma cell proliferation, migration, and invasion [162]. Taken together, these findings suggest that MEG3 may function as a prognostic marker and a potential target for glioblastoma therapy.

##### *HOTAIR* 

Homo sapiens HOX transcript antisense RNA (*HOTAIR*) is a human intergenic lncRNA located within the Homeobox C (HOXC) gene cluster on chromosome 12, being co-expressed with the *HOXC* genes. This lncRNA is highly expressed in several types of tumors and some splicing variants have been described [163]. Mechanistically, the 5′ end of HOTAIR binds to the Polycomb Repressive Complex 2 (PRC2) resulting in regulation of the chromatin state and promoting epigenetic repression of the *HOXD* locus by PRC2 [164]. The 3′ end of HOTAIR interacts with the Histone Demethylase (LSD1), which is involved in demethylation of histone H3 at lysine 4, leading to gene expression control [165]. Similarly to MALAT1, HOTAIR also functions as a sponge miRNA to regulate gene expression in cancer. HOTAIR acts as a ceRNA to regulate *HER2* expression by sponging miR-331-3p [166] and reprogramming the chromatin state to promote cancer metastasis [167,168]. Tan et al. detected HOTAIR in serum from glioblastoma patients [169]. The levels in serum samples from glioblastoma patients was significantly higher when compared with those of control healthy subjects, demonstrating that HOTAIR may be used as a prognostic biomarker for glioblastoma [169,170]. HOTAIR depletion acts as an anti-cancer agent in glioblastoma, regulating the FGF1-dependent pathway through miR-326 [171]. Additionally, *HOTAIR* knockdown inhibited glioblastoma formation in vivo, indicating that this lncRNA was required for tumor development [172]. HOTAIR and the HOTAIR/miR-326/FGF1 axis are a promising therapeutic strategy for glioblastoma therapy.

##### *H19* 

H19 imprinted maternally expressed transcript (*H19*), is a lncRNA whose coding gene is located on an imprinted region of human chromosome 11, adjacent to the insulin-like growth factor 2 (*IGF2*) gene [173]. The product of the *H19* gene is a 2.3 kb spliced, capped and polyadenylated lncRNA [174]. H19 is required for gene expression control by imprinting regulation, including *IGF2* [175]. Multiple biological roles have been attributed to H19 and functional studies classify it as an oncogene [176,177]. The hypoxia microenvironment induces endogenous expression of *H19* via Hif-1α, which directly binds to the *H19* promoter, triggering the oncogenic effects in tumor cells [178]. The Hif-1α stability is attenuated by *PTEN*, a tumor suppressor gene, demonstrating that Hif-1α positively correlates with H19; however, this mechanism is dependent on the status of *PTEN* [178]. H19 also functions as ceRNA to regulate epithelial-to-mesenchymal transition by sponging miR-130a-3p, a critical biological process, which leads to polarization, migration, adhesion, and invasion in glioma cells [179]. Additionally, studies demonstrated that *H19* is overexpressed in glioblastoma and that this pattern promotes glioma cell invasion by activation of miR-675 [180]. Other studies have confirmed the functional impact of H19 in the context of brain tumor. H19 plays an important role in glioblastoma tumorigenicity, since its overexpression promotes invasion, angiogenesis, stemness and increased tumor growth in vivo [181]. Finally, the potential tumorigenicity of glioblastoma cells may also be affected by recruitment of EZH2 (a subunit of the Polycomb Repressive Complex 2) to the *NKD1* promoter by H19, with its repression positively contributing to glioblastoma tumorigenicity [182].

##### *NEAT1* 

Nuclear paraspeckles assembly transcript 1 (*NEAT1*) is a lncRNA transcribed from the multiple endocrine neoplasia locus located on human chromosome 11 [183]. Once *NEAT1* is retained in the nucleus, it adopts an architectural function, associating with specific proteins (e.g., P54nrb or NONO), forming a structural core called paraspeckles [184]. NEAT1 functions as a scaffold RNA by interacting with different target genes [185] and acts on transcriptional activation of various genes associated with cancer, including glioblastoma [186]. The interaction between NEAT1, miR-132, and *SOX2* has an important role in glioblastoma. NEAT1 indirectly regulates *SOX2* expression by sponging miR-132. The NEAT1/miR-132/SOX2 axis regulates glioblastoma cells invasiveness, and, therefore, could be explored for treatment of this tumor type [186]. NEAT1 depletion inhibits GSCs migration and invasion and promotes cell death by activating the miRNA let-7e [187]. Systematic analysis of public glioma expression data sets showed that *NEAT1* is an oncogene [188]. Moreover, Chen et al. demonstrated that *NEAT1* expression was regulated by the EGFR pathway mediated by nuclear translocation of STAT3 and NF-kB [188]. Additionally, NEAT1 triggers β-catenin translocation, which leads to downregulation of *ICAT*, *GSK3B*, and *Axin2*. Mechanistically, NEAT1 binds to EZH2, mediating trimethylation of the H3K27 present in their promoters, leading to transcriptional repression of these genes [188]. *NEAT1* knockout mediated by the CRISPR/Cas9 system showed that intracranial tumor growth and invasion was abolished in vivo [188]. In summary, the EGFR/NEAT1/EZH2/β-catenin axis plays a crucial role in the development of glioblastoma. 

##### *XIST* 

X-inactive specific transcript (*XIST*) is a ncRNA expressed from the X chromosome of placental mammals, acting directly in X chromosome inactivation [189], which is widely known to occur during the developmental process in mammalian females by silencing one of the X chromosome through a mechanism known as “dosage compensation” between males and females [190]. The X Inactivation Center (XIC), located in the X chromosome, is responsible for regulation of this process, comprising several lncRNAs, including *XIST*. This transcript is a spliced RNA, of 17kb length in humans, being expressed solely from the XIC of the inactive X chromosome by a *cis*-regulatory mechanism [191]. *XIST* RNA is upregulated in glioblastoma cells and its depletion inhibits glioblastoma angiogenesis by Zonula Occludens 2 (ZO-2) and Forkhead Box C1 (FOXC1) transcriptional inactivation [192]. Both ZO-2 and FOXC1 are crucial for maintaining the blood–tumor barrier integrity through upregulation of miR-137 [192]. *XIST* RNA also acts as a molecular sponge for miR-429 in glioblastoma cells and the negative modulation of *XIST* contributes to repression of glioblastoma cells metastatic and angiogenic potential [193]. On the other hand, *XIST* depletion exerts tumor-suppressive roles in GSCs by upregulating miR-152 [194].

#### 4.2.2. Tumor Suppressor LncRNAs

##### *RAMP2-AS1* 

RAMP2-AS1 is a long ncRNA whose coding gene is located on human chromosome 17 and the product of this gene is a 1.9kb spliced and polyadenylated lncRNA. RAMP2-AS1 was initially identified as a ncRNA which is differentially expressed between adenocarcinoma and squamous cell carcinoma [198]. Shuang Liu et al showed that *RAMP2-AS1* is downregulated in glioblastoma tissues and its ectopic overexpression reduces tumor growth in a subcutaneous mouse model [140]. This lncRNA interacts with the DHC10/NOTCH3/HES1-signaling pathway and plays a tumor-suppressive role in glioblastoma progression through inhibition of NOTCH3 [140]. The lncRNA–mRNA interaction provides insights into the effects of this ncRNA in glioblastoma pathogenesis. 

##### *CASC2* 

Cancer Susceptibility Candidate 2 (*CASC2*) is located on human chromosome 10 and the product of this gene is a 3.3kb lncRNA. *CASC2* was originally identified in endothelial cancer, acting as a potential tumor suppressor gene [199]. One of the products of this gene is *CASC2a*, which encodes a small protein of 102 amino acids, whose function depends on the genetic alterations present in this gene [199]. In glioma tissues and glioblastoma cell lines, the *CASC2* lncRNA is downregulated [200]. Overexpression of *CASC2* inhibited malignancy of glioblastoma cells via miR-21, and overexpression of *miR-21* abrogated the *CASC2*-induced inhibitory effects in the same cells [200]. Mechanistically, miR-21 binds to *CASC2* in a sequence-specific manner, leading to transcriptional repression of *CASC2*. Consequently, this process triggers glioblastoma cells proliferation, migration, and invasion [200]. Similarly, overexpression of *CASC2* also acts by suppressing the Wnt/β-catenin signaling pathway in glioblastoma cells, leading to suppression of cellular events similarly to those previously described [201]. Glioblastoma cells display resistance to TMZ by mechanisms related to increased autophagy [202]. Interestingly, studies show that sensitivity to TMZ is CASC2-dependent in GSCs. *CASC2*-overexpression inhibits autophagy and increases the susceptibility of GSCs to TMZ treatment by accumulation of lipid peroxides (ferroptosis), leading to cell death [202]. It has recently been reported that the mechanism of autophagy inhibition, mediated by upregulation of *CASC2*, occurs by sponging miR-193a-5p and regulating *mTOR* expression [203]. In the presence of TMZ, the CASC2/miR-193a-5p/mTOR axis could be a promising therapeutic approach for glioblastoma treatment.

Other lncRNAs have been linked with aggressiveness or as predictors of survival in glioblastoma patients, such as *GAS5* [204], *TP73-AS1* [138], and *CRNDE* [195]. However, more efforts are needed to better understand the molecular implications of these lncRNAs in glioblastoma pathogenesis. 

## 5. Innovative Clinical Applications of ncRNAs for Glioblastoma

Following the advent of RNA-sequencing technologies, our understanding of the molecular etiology of diseases has drastically increased. Our ability to detect and better discern the biological heterogeneity present in tumors has revealed that each tumor has a particular identity, so that personalized/precision medicine represents a promising future strategy for patient treatment. In line with these advances, the 2016 WHO classification of diffuse gliomas has incorporated molecular tests allowing for more precise diagnosis and providing guidance for therapeutic interventions. Currently, molecular tests are routinely used in the clinic as requisite for glioma categorization, such as the identification of isocitrate dehydrogenase (*IDH1/2*) mutation, 1p/19q co-deletion, H3 Histone Family Member 3A (*H3F3A*) mutation, gene for Histone (*H3HIST1H3B/C*) mutation and *C11orf95–RELA* fusions.

A growing number of studies have shown that the ncRNA signature in glioblastoma predicts patient survival and that several ncRNAs may be used as diagnostic biomarkers or even serve as potential therapeutic targets. For instance, circulating miRNAs levels, such as miR-221/222 and miR-137, have emerged as potential non-invasive biomarkers for diagnosis and tracking cancer progression in patients with gliomas. LncRNAs can also be easily isolated from circulating cells, readily providing results and thus allowing inexpensive blood-borne diagnostics for more efficient detection of cancers. 

Moreover, different miRNAs and lncRNAs were shown to affect chemo- and radio-sensitivity and RNA-based therapy could provide a complementary strategy for more effective treatment against chemo- and radioresistant glioblastomas. Notably, the promising results obtained with mir-10b targeting therapy gave support to the ongoing clinical trial #NCT01849952 expected to be completed in 2022. This study aims to test *mir-10b* expression patterns as a prognostic and diagnostic marker in gliomas and to investigate the in vitro sensitivity of individual primary tumors to anti-mir-10b treatment. Such evaluation and validation in larger-scale prospective studies are required to facilitate the development of novel tools and support implementation of these strategies in routine clinical practice.

## 6. Concluding Remarks

LncRNAs play a central role in transcriptional and post-transcriptional regulation of protein-coding genes and may be categorized into different archetypes, such as: ceRNAs/miRNA sponges, guides, scaffolds, or enhancers. MiRNAs act at the post-transcriptional level by mRNA cleavage, blocking mRNA translation and/or mRNA stability. 

LncRNAs and miRNAs are critical ncRNAs inserted in a complex regulatory network, with abnormal expression of these molecules having a direct impact on several aspects of gliomagenesis.

Several studies have demonstrated the potential application of these molecules as predictors for clinical prognosis/diagnosis or as therapeutic targets. The intrinsic genetic heterogeneity of glioblastoma specimens represents a great challenge in the field, therefore, large-scale clinical trials are required to validate the practical value of these findings to the clinic.

In conclusion, the lncRNA–miRNA–mRNA crosstalk represents a master regulatory key for maintenance of cellular homeostasis. The lncRNA-miRNA co-expression network provides an extra layer of complexity into how these molecules can contribute to glioblastoma onset, progression, and maintenance. This complex network of lncRNA-miRNA interactions is a prominent field of research, which may reveal potential therapeutic options for patient-tailored treatment.

## Figures and Tables

**Figure 1 ijms-21-02611-f001:**
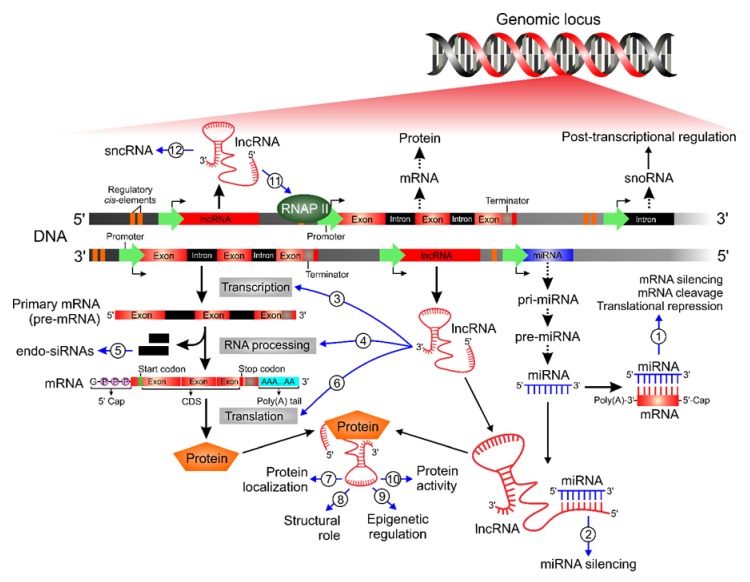
Schematic overview of ncRNA biogenesis, functions, and mechanisms of action in glioblastoma cells. This scheme represents the complexity of a genomic locus and the main molecular events which contribute to gene expression control. LncRNAs and miRNAs are master regulators in multiple biological processes associated with the initiation and progression of glioblastoma. (**1**) miRNAs directly interact with the target mRNAs, leading to mRNA silencing, mRNA cleavage or repression of protein translational. LncRNAs can be categorized according to their cellular functions and mechanisms of action: (**2**) acts as ceRNA by sponging miRNA; (**3**) direct transcriptional control; (**4**) RNA processing, including mRNA stability and alternative splicing; (**5**) endo-siRNAs production by spliced intron; and (**6**) protein translation regulation. Moreover, lncRNAs facilitate the assembly of ribonucleoprotein (RNP) complex which can affect different processes, such as (**7**) subcellular localization of proteins, (**8**) structural and organizational roles, (**9**) remodeling of chromatin, and (**10**) post-translational regulation of protein activity. Additionally, lncRNAs contribute to (**11**) activate or suppress RNAP II activity and can be used to (**12**) produce sncRNAs, which have regulatory/structural functions. RNAP II: RNA Polymerase II; CDS: coding sequence; lncRNA: long non-coding RNA; miRNA: microRNA; endo-siRNA: endogenous small interfering RNA; sncRNA: small non-coding RNA; snoRNA: small nucleolar RNA.

**Table 1 ijms-21-02611-t001:** Summary of selected oncogenic miRNAs with their functional effects in glioblastoma.

Major Oncogenic miRNAs in Glioblastoma			
MicroRNA	Validated Targets	Functional Effects in Glioblastoma	References
hsa-miR-21	IGFBP3, RECK, TIMP3, ANP32A, Bcl-2, PTEN, HNRPK, TAP63, MSH2 LRRFIP1, PDCD4 (…)	Promotes cell proliferation, invasion, chemoresistance and tumor growth in vivo	[60,61,63]
hsa-miR-26a	PTEN	Enhances tumor formation in vivo	[85]
hsa-miR-19a/b	RUNX3, CTGF	Promotes cell proliferation and G1 cell cycle progression; modulates cell apoptosis and invasion	[86,87]
hsa-miR-93	Integrin-β8, P21	Promotes cell proliferation, cell cycle progression, migration, invasion, and chemoresistance; induces angiogenesis and enhances tumor growth in vivo	[88,89,90]
hsa-miR-221/222	P27, AKT, PUMA, P57, PTPμ, Cx43, TIMP3, MGMT	Promotes cell proliferation, invasion, and chemoresistance; modulates cell apoptosis and tumor growth in vivo	[75,77,78,79,80,81,82,83]
hsa-miR-20a	TGFβ-RII, CTGF, CELF2, LRIG1	Promotes cell proliferation, invasion and inhibits apoptosis	[86,91,92]
hsa-miR-25	Mdm2, TSC1, P57, NEFL	Promotes cell proliferation, invasion and cell cycle progression	[93,94]
hsa-miR-130b	CYLD	Promotes cell proliferation, invasion and inhibits apoptosis	[95,96]
hsa-miR-210	HIF3α, SIN3A	Promotes cell proliferation and inhibits cell apoptosis; mediates hypoxic survival and enhances chemoresistance	[97,98]
hsa-miR-155	GABRA-1, FOXO3a, MXI1, MAPK13/14	Promotes cell proliferation and invasion and inhibits apoptosis	[94,99,100,101]
hsa-miR-10b	PTEN, BIM, P21, P16, TFAP2C, MBNL2, MBNL3, SART3, HOXD10	Promotes cell proliferation, cell cycle progression, migration, invasion, and inhibits apoptosis; modulates tumor growth in vivo	[69,70,71,73,102,103,104]

**Table 2 ijms-21-02611-t002:** Summary of selected tumor suppressor miRNAs with their functional effects in glioblastoma.

Major Tumor Suppressor miRNAs in Glioblastoma			
MicroRNA	Validated Targets	Functional Effects in Glioblastoma	References
hsa-mir-34a	SIRT1, c-Met, Notch1/2, PDGFRA, Msi1	Inhibits cell proliferation, cell cycle progression, cell survival, invasion, and tumor growth in vivo	[125,126]
hsa-miR-128	WEE1, p70S6K1, Msi1, E2F3a, SUZ12, Bmi-1, EGFR, PDGFRα, ANGPTL6	Decreases radioresistance, attenuates cell proliferation, tumor growth and angiogenesis	[105,108,110,111,112,127]
hsa-miR-137	RTVP-1, COX-2, EGFR, CDK6, RTVP-1 Rac1	Inhibits proliferation and invasion and reduces stemness; increases apoptosis and promotes cell cycle arrest	[120,121,124,128]
hsa-miR-124	AURKA, SOS1	Inhibits proliferation, reduces stemness, promotes cell cycle arrest and increases chemosensitivity.	[121,129,130]
hsa-miR136	AEG-1, Bcl-2	Promotes apoptosis and increases chemosensitivity	[131,132]
hsa-miR-181b	FOS, MEK1, IGF-1R, CCL8, MDM2	Inhibits proliferation, migration, and invasion; promotes cell cycle arrest; suppresses angiogenesis and tumor growth in vivo	[114,116,117,119,133,134]
hsa-miR-195	E2F3, CCND3, Cyclin D1, Cyclin E1	Inhibits proliferation, migration, and invasion; promotes cell cycle arrest and reduces tumor growth in vivo	[135,136]
hsa-miR-139-5p	ELTD1, Notch1	Inhibits proliferation and invasion; promotes apoptosis; reduces tumor growth and prolongs survival in vivo	[103,104]

**Table 3 ijms-21-02611-t003:** Summary of selected oncogenic lncRNAs with their potential roles in glioblastoma.

Major Oncogenic LncRNAs in Glioblastoma				
LncRNA	Target miRNA	Mechanism of Action	Functional Effects in Glioblastoma	References
MALAT1	miR-199a miR-203	Acts as a molecular sponge for miRNAs	Promotes cell proliferation and tumorigenesis; leads to resistance to TMZ-treatment	[144,152]
MEG3	miR-19a	Acts as a ceRNA for miRNAs, represses *PTEN* expression and controls the expression of p53-target genes	Increases cell proliferation, migration, and invasion	[162]
HOTAIR	miR-326	Binds to EZH2 and regulates FGF1-dependent pathway by acting as a sponge for miRNA	Promotes cell proliferation and glioblastoma cells growth	[171]
H19	miR-675 miR-130a-3p	Binds to EZH2 and acts as a molecular sponge for miRNAs	Promotes invasion, angiogenesis, stemness and increased glioblastoma cells growth	[179,180]
NEAT1	miR-132 let-7e	Binds to EZH2, functions as a scaffold RNA by interacting with target genes and triggers β-catenin translocation	Promotes tumor progression, regulates invasiveness of glioblastoma cells and promotes GSCs migration and invasion	[186,187]
XIST	miR-137 miR-429 miR-152	Acts as a ceRNA for miRNAs and promotes transcriptional inactivation of ZO-2 and FOXC1	Promotes angiogenesis and has a potential role in GSCs	[192,193,194]
TP73-AS1	-	Is linked to reduced *ALDH1A1* expression	Promotes tumor aggressiveness and TMZ resistance in GSCs; prognostic biomarker	[138]
CRNDE	miR-186 miR-136-5p	Negatively regulates miRNAs	Promotes cell growth and GSCs proliferation; is a prognostic factor for glioblastoma patients	[195,196,197]

**Table 4 ijms-21-02611-t004:** Summary of selected tumor suppressor lncRNAs with their potential roles in glioblastoma.

Major Tumor Suppressor LncRNAs in Glioblastoma				
LncRNA	Target miRNA	Mechanism of Action	Functional Effects in Glioblastoma	References
RAMP2-AS1	-	Interacts with DHC10/NOTCH3/HES1-signaling pathway	Reduces tumor growth	[140]
CASC2	miR-21 miR-193a-5p	Binds directly to miRNA, suppresses the Wnt/β-catenin signaling pathway and regulates *mTOR* expression	Inhibits autophagy and malignancy in glioblastoma cells, sensitizes GSCs to TMZ-treatment leading to ferroptosis	[200,203]
GAS5	miR-222 miR-196a-5p miR-18a-5p	Acts as a molecular sponge for miRNAs	Promotes proliferation in glioblastoma cells and GSCs; prognostic predictor of survival	[204,205,206]

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
