# Peer review of "Emerging Roles and Potential Applications of Non-Coding RNAs in Glioblastoma"

_ijms, 2020, doi:10.3390/ijms21072611_

Round 1
Reviewer 1 Report
This manuscript reviews molecular mechanisms and applications of miRNA and lncRNA in glioblastoma.
In the title, ‘potential’ is unclear. It needs to be described in a concrete manner.
It is recommended to describe more details for (i) – (v), lines 142 – 149, p4.
More references are required for miRNA deregulation in section 4.1 and this section needs to re-written.
Describe more therapeutic applications especially in concluding remarks with your ideas or in other sections.
Author Response
please see the attachment below, thanks.

Reviewer 2 Report
This review article submission by DeOcesano-Pereira et al explore the potential role and applications of non-coding RNAs in glioblastoma.
I found this review very interesting, well written and nicely articulated. It summarizes nicely the different type of ncRNA and also what is known about their role in GBM using different model systems. The review summarizes also nicely the role of some specific ncRNA in brain cancer and potential therapeutic interventions. Overall, it falls well within the scope of the journal and I would recommend its publication in it.
I would just made a small comment regarding figures. I think TCGA-derived survival plots from the different ncRNA that were described in the text would help readers to visualize the impact that ncRNA expression has on survival as well as illustrate some of the concepts that were highlighted in some passages of the text.
Author Response

(The authors gave the same response as above.)

Reviewer 3 Report
The review presents a good and appropriate introduction to non-coding RNAs in general, and then presents the known non-coding RNAs related to glioblastoma.
I think the most important contribution of a review article is to present information and views that a novice may not easily get from reading the primary articles. In this vein, I have a few small suggestions that could make the review even more useful. These are mostly related to tying together the information and making meaningful conclusions.
- With regard to how the non-coding RNAs and their phenotypes are discovered, what is the authors' view on and biases and blind spots? eg, are we more likely to have discovered tumor suppressors or oncogenes due to how the experiments are designed or processed? How likely are we to discover new glioblastoma related non-coding RNAs? Is this a problem similar to protein coding gene phenotype discovery or unlike it? Are the non-coding genes more likely to be causal and part of the initiation of the tumor, or do they have a role in the subsequent effects of tumor growth and progression.
- Are any of the ncRNAs currently being used as biomarkers for diagnosis and prognosis in the clinic, or being developed for such?
- It might also be helpful if tables 1 and 2 are also presented as a network of lncRNAs -> mRNAs -> protein coding gene -> tumor suppressor/oncogene effect. The network picture might be particularly helpful in framing new hypotheses to test.
Author Response

(The authors gave the same response as above.)

Reviewer 4 Report
The review article describes the role of non-coding RNAs in the regulation of glioblastoma.
The article covers sufficient details.
Minor comments:
Some of the fonts are larger in a few a paragraphs. The authors should update for consistency.
Author Response

(The authors gave the same response as above.)
